# Efficacy and Synergistic Potential of Cinnamon (*Cinnamomum zeylanicum*) and Clove (*Syzygium aromaticum* L. Merr. & Perry) Essential Oils to Control Food-Borne Pathogens in Fresh-Cut Fruits

**DOI:** 10.3390/antibiotics13040319

**Published:** 2024-03-31

**Authors:** Ramona Iseppi, Eleonora Truzzi, Carla Sabia, Patrizia Messi

**Affiliations:** 1Department of Life Sciences, University of Modena and Reggio Emilia, Via G. Campi 287, 41125 Modena, Italy; ramona.iseppi@unimore.it (R.I.); carla.sabia@unimore.it (C.S.); 2Department of Chemical and Geological Sciences, University of Modena and Reggio Emilia, Via G. Campi 103, 41125 Modena, Italy; eleonora.truzzi@unimore.it

**Keywords:** cinnamon and clove essential oils, synergistic activity, food-borne pathogens, fresh-cut fruit, bio-preservation

## Abstract

The presence of microbial pathogens in ready-to-eat produce represents a serious health problem. The antibacterial activity of cinnamon (*Cinnamomum zeylanicum*) and clove (*Syzygium aromaticum* L. Merr. & Perry) essential oils (EOs) was determined toward food-borne pathogens by agar disk diffusion and minimum inhibitory concentration (MIC) assays. The growth kinetics of all strains, both in a buffer suspension assay and “on food” in artificially contaminated samples, were also investigated. The two EOs demonstrated a good antibacterial effect both alone and in combination (EO/EO). The use of EO/EO led to a synergistic antibacterial effect, also confirmed by the growth kinetics studies, where the EOs were active after 10 h of incubation (*p* < 0.0001) at significantly lower concentrations than those when alone. In the “on food” studies performed on artificially contaminated fruit samples stored at 4 °C for 8 days, the greatest killing activity was observed at the end of the trial (8 days) with a reduction of up to 7 log CFU/g compared to the control. These results confirm the good antibacterial activity of the EOs, which were more effective when used in combination. Data from the "on food" studies suggest cinnamon and clove essential oils, traditionally used in the food industry, as a possible natural alternative to chemical additives.

## 1. Introduction

The Mediterranean diet is known throughout the world for its balanced nutrient content, and in this context, ready-to-eat (RTE) fruits and vegetables offer a range of minerals, vitamins and phytochemicals essential for human health [1]. Adequate consumption of fresh produce plays an important role in the prevention of chronic pathologies such as cardiovascular diseases, cancer, hypertension, diabetes and obesity [2]. Given that the beneficial health value of these foods is widely recognized by the scientific community and consumers, the ready-to-eat (RTE) fresh produce market is one of the most rapidly expanding sectors of the food industry [3].

It is well known that the mild treatments undergone by RTE produce favor more rapid physiological deterioration, biochemical changes and microbial degradation, mainly in fruits for their high water and sugar contents [4]. This can cause both economic loss through spoilage bacteria and lead to outbreaks, resulting in illness and death due to the presence of food-borne pathogens. Among RTE produce, fresh-cut fruits could act as a vehicle for transmitting pathogens of major concern like *Listeria monocytogenes*, *Staphylococcus aureus*, pathogenic *Escherichia coli* mainly O157:H7 and *Salmonella* spp. [5,6]. 

Several listeriosis outbreaks have been linked to fresh produce contamination around the world, and many studies have detected *L. monocytogenes* in fresh minimally processed fruits [7,8,9] linked to melons, caramel apples, peaches, plums and nectarines (stone fruits) [10,11,12].

*S. aureus* has been detected on fresh produce and ready-to-eat vegetable salads, and it proved capable of growing on peeled Hamlin oranges stored at 24 °C (75.2 °F) and survived up to 14 days when stored at 4 to 8 °C (39.2 to 46.4 °F) [13]. 

*E. coli* O157:H7 is recognized as an important food-borne pathogen, and it has been identified as a causative agent for cantaloupe outbreaks in 1993, 1997 and 2004 [14,15]. 

Multistate outbreaks of *Salmonella enteritidis* and *Salmonella* Sundsvall linked to the consumption of peaches and imported Mexican cantaloupes, respectively, were reported by the U.S. Food and Drug Administration (FDA) in 2021 and 2024 [16,17], and a *Salmonella* Newport outbreak in ready-to-eat (RTE) cut watermelons was reported in the United Kingdom (UK) in 2012 [18].

*Y. enterocolitica* is a ubiquitous pathogen frequently isolated from soil and water from fruit and vegetables, thus being a potential source of yersiniosis [19,20]. In 2021, berries and juices and other derived produce were implicated in outbreaks caused by unspecified *Salmonella* and *Y. enterocolitica* [21].

Fresh-cut produce, due to its high water content, can favor the development of pathogens and spoilage bacteria [22]. To counteract their multiplication in food, chemical preservatives have always been used, but consumer demand is increasingly oriented towards a safe product with a long shelf-life obtained with the use of natural preservatives. Recent exploitation of natural products to control decay and extend the shelf-life of perishables has received more and more attention by researchers, including essential oils (EOs), natural compounds isolated from aromatic plants and generally recognized as safe (GRAS) for the environment and human health. Although originally added to change or improve taste, essential oils have antimicrobial activity that makes them an attractive choice to replace synthetic preservatives [23,24]. Many studies have demonstrated the antimicrobial activity of several spices toward pathogenic and spoilage bacteria and fungi, such as cinnamon and clove. *Cinnamomum zeylanicum* (cinnamon) is a very common spice obtained from the bark and leaves of trees of the *Cinnamomum* genus, widely used as a flavoring component in the food industry. The main components of cinnamon EO extracted from bark are *trans*-cinnamaldehyde, eugenol and linalool. Based on several studies, cinnamon essential oil exhibits significant inhibitory effects on both Gram-negative and Gram-positive bacteria like *Salmonella* enterica, *E. coli*, *S. aureus* and *L. monocytogenes* [25]. *Syzygium aromaticum* (clove) EO is considered the most useful and beneficial among spices that have been used in the food field. Clove contains flavonoids and phenolic molecules, including eugenol, which represents the main bioactive molecule of this EO [26]. With its main component of eugenol, clove EO has high potential not only for pharmaceutical applications but also for agricultural and food applications. The antimicrobial activities of clove EO appear to surpass those of other spices, showing extraordinary antibacterial activity against a great number of microorganisms [27].

The use of essential oils in the food industry is an interesting approach, but their intense aroma limits their use in the field of food preservation. An approach to overcome this problem could be to create combinations of different essential oils that, due to their synergistic effects, could allow reducing the concentration and, consequently, the negative sensorial impact without compromising the antimicrobial activity [28].

The aim of the present investigation was to assess the efficacy and synergistic potential of two essential oils traditionally used in the food industry to control food-borne pathogens in fresh-cut fruits. In this study, cinnamon and clove essential oils (EOs) were used alone and in combination to find a safe, effective and natural method to improve the safety and the shelf life of highly perishable produce.

## 2. Results

### 2.1. Chemical Composition of the EOs

*C. zeylanicum* (cinnamon) and *S. aromaticum* EOs were phytochemically characterized in terms of chemical composition via GC–MS and GC–FID. More than 95% of the total composition was determined for each EO. As reported in Table 1, cinnamon EO was mainly rich in *trans*-cinnamaldehyde, which accounted for 69% of the total composition. Relevant amounts of β-caryophyllene, linalool, eugenol, limonene, cinnamyl acetate, γ-cadinene, *p*-cymene and α-humulene were also detected. Regarding clove EO, eugenol and its ester derivative represented more than 91% of the total composition. 

### 2.2. Agar Disk Diffusion Assay

Table 2 shows the antibacterial activity of cinnamon and clove EOs using the agar disk diffusion assay as a preliminary screening. Both Gram-negative and Gram-positive bacteria were sensitive to the EOs. Cinnamon and clove EOs displayed a bacterial growth inhibition zone ranging from 27 to 32 mm and 30 to 35 mm, respectively, with similar inhibition zones observed for both Gram-positive and Gram-negative strains.

### 2.3. Minimum Inhibitory Concentration (MIC) and Fractional Inhibitory Concentration Index (FIC Index) Determination

The results of the minimum inhibitory concentration (MIC) of both EOs, performed on the same strains, confirmed the good antibacterial activity against both Gram-positive and Gram-negative microorganisms, as revealed by the disk diffusion test (Table 3). Cinnamon EO showed an MIC value ranging from 1 to 8 μg/mL, whereas clove EO displayed a higher MIC value (from 4 to 64 μg/mL), especially toward *S. typhimurium* and *L. monocytogenes* (64 μg/mL for both strains).

Regarding the effectiveness of EOs used in combination (cinnamon EO/clove EO), a synergistic effect against all tested strains was observed. The combination of EOs led to an increase in the antibacterial effect, with a consequent decrease in the concentrations used (four times) (Table 3).

### 2.4. Growth Kinetics Study

Cinnamon and clove EOs showed activity against viable cells of all tested strains after 10 h of incubation, especially for *E. coli* (Appendix A). Both EOs and the EO/EO combination were active (*p* < 0.0001) at the end of the experiments toward Gram-negative and Gram-positive microorganisms, notably against *L. monocytogenes* (Figure 1). Also, in this case, the mixture of EOs allowed a significant reduction in the concentrations of the individual EOs.

### 2.5. Evaluation of the Antibacterial Activity by “On Food” Studies

The trend observed for *Y. enterocolitica* ATCC 23715 in fresh-cut fruits added with cinnamon and clove EOs, alone or in combination, and stored at refrigeration temperature showed a viable cell decrease after 24 h. After 4 days, for both the cinnamon and clove EOs and the EO mixture, a further reduction of about 2.00 log CFU/g compared to the control (*p* = 0.0007, *p* = 0.0001 and *p* = 0.0002, respectively) was observed. After 6 days, both EOs resulted in a reduction of approximately 5 log CFU/g (*p* < 0.0007 for cinnamon and clove EOs), and the EO/EO combination displayed a reduction of 6 log CFU/g (*p* < 0.0001). At the end of the trial (8 days), cinnamon and clove EOs showed a viable cell reduction of 6 log CFU/g and 6.5 log CFU/g, respectively (*p* < 0.0001), and for the EO/EO combination, a 7 log CFU/g decrease was observed (*p* < 0.0001) (Figure 2).

Regarding *E. coli* ATCC 25922, both single EOs and the EO/EO combination showed a reduction in viable cells of about 2 log CFU/g (*p* < 0.01) after 24 h of the experiment. After 4 days, clove EO and the EO/EO combination displayed the best results (*p* < 0.0001) compared to the control with a reduction of about 5 log CFU/g, and the cinnamon EO showed a reduction of 4 log CFU/g (*p* < 0.0001). After 6 days of the trial, clove EO and the combination still confirmed the best results with a reduction of 6 log CFU/g (*p* < 0.0001) of viable *E. coli* cells. At the end of the trial (8 days), the EO/EO combination had the best results, showing 7.7 log CFU/g *E. coli* viable cells (Figure 3).

Regarding *S. typhimurium* ATCC 19585, at 24 h of the experiment, both EOs displayed a reduction of about 1.5 log CFU/g in viable cells, and a better result emerged using the EO/EO combination, with a reduction of 2.5 log CFU/g viable cells (*p* = 0.0007). After 4 days, cinnamon and clove EOs alone and the EO/EO combination exhibited the same activity with a reduction of about 4 log CFU/g (*p* < 0.001) viable cells. From day 6 until the end of the experiment, clove EO and the EO/EO combination demonstrated the best activity against *S. typhimurium* compared to the control (*p* < 0.0001) (Figure 4).

Figure 5 shows the anti-listeria activity of cinnamon and clove EOs and of the EO/EO combination throughout the experiment time (8 days). The single EOs and the EO/EO combination displayed the same activity after 24 h (*p* < 0.01), and after 4 days, the EO/EO combination showed the best activity with a reduction of about 4.5 log CFU/g compared to the control (*p* = 0.0002). From day 6 until the end of the experiment, all samples (EOs alone and the EO/EO combination) demonstrated remarkable activity against *L. monocytogenes* compared to the control (*p* < 0.0001).

Concerning *S. aureus*, both the single EOs showed a reduction in viable cells of about 1.50 log CFU/g (*p* = 0.006) after 24 h of the experiment, and the EO/EO combination displayed a reduction of 2.5 log CFU/g (*p* = 0.0011). After 4 days, clove EO demonstrated the best results (*p* = 0.0002) compared to the control with a reduction of about 4 log CFU/g, and the cinnamon EO and the EO/EO combination showed a reduction of 3.7 log CFU/g (*p* = 0.0005). Clove EO and the EO/EO combination showed the best results with a reduction of about 5.5 log CFU/g (*p* < 0.0001) viable *S. aureus* cells, even after 6 days of the experiment. At the end of the trial (8 days), the EO/EO combination presented the best result with a 7.6 log CFU/g reduction of *S. aureus* viable cells (Figure 6).

## 3. Discussion

Today’s busy lifestyles negatively affect eating habits, and the limited time to prepare meals, even simple ones, creates the necessity to purchase ready-to-eat produce that constitutes a suitable meal, as it does not require extra preparation, is easy to consume and represents a valid alternative to maintain a balanced diet [30]. Natural barriers such as skin and rinds as well as a naturally acidic pH prevent or retard the growth of pathogenic bacteria in fruits. However, some fully matured fruits have pH values approaching 7, and once cut, they expose the internal flesh to environmental contaminants and can serve as substrates for the growth of bacteria [14]. Cantaloupe and watermelon are among these fruits and are not exempt from the list of foods known to be vehicles of food-borne illness [31]. A critical aspect of RTE foods like this is microbial contamination, and the primary objective of the manufacturer must be not only to obtain an increasingly prolonged shelf life but also to guarantee the safety of the product. In the present investigation, the EOs showed antimicrobial efficacy against all the food-borne pathogens, both alone and in combination. Clove EO has already shown its effective antimicrobial efficacy against some food-borne pathogens [32]. The main active component of clove EO is eugenol [33,34], which can deteriorate the cell wall and determine cell lysis [35]. *Trans*-cinnamaldehyde is the main compound in cinnamon EO [36]. Cinnamaldehyde has antimicrobial effects, as it inhibits cell wall biosynthesis, membrane function and certain enzymatic activities [37]. Also, in the present investigation, clove and cinnamon EOs were found to be abundantly effective against both Gram-negative and Gram-positive strains. Clove EO and cinnamon EO showed the same anti-listeria activity throughout the experiment. The single EOs displayed the same activity against *Y. enterocolitica* and *S. typhimurium*, although clove was more effective than cinnamon at 4 and 6 days of experimentation for *Yersinia* and at 6 days for *Salmonella*. Concerning *E. coli* and *S. aureus*, clove EO exhibited better activity than cinnamon EO at 4 and 6 days of experimentation. Clove EO reduced viable cells of *E. coli* by 5 log and 6 log at 4 and 6 days, whereas cinnamon EO showed a reduction of 4 log and 5 log, respectively. Other authors have demonstrated the better activity of clove EO compared to other essential oils such as cinnamon, cardamom and oregano against Gram-negative and Gram-positive bacteria such as *S. typhimurium*, *E. coli*, *Bacillus cereus* and *Listeria innocua* [38,39]. Furthermore, Liang et al. [40] demonstrated that clove EO has better antimicrobial activities against spoilage microorganisms in apple cider than those of other spices tested. Other studies have also highlighted the antibacterial activity of cinnamon EO in food-borne and spoilage bacteria such as *B. cereus*, *E. coli*, *Salmonella enterica*, *Y. enterocolitica*, *L. monocytogenes*, *S. aureus* and *P. aeruginosa* [41,42]. Cinnamon EO displays prominent activity against fungi that are more sensitive compared to bacteria [26,43]. Moreover, cinnamon EO exhibits antibacterial and anti-biofilm activity against methicillin-resistant *S. aureus* (MRSA) [44]. *S. aureus* can acquire antimicrobial resistance, complicating the problem of food-borne illnesses. Furthermore, ready-to-eat (RTE) foods, which do not require heat treatment before consumption, represent a vehicle for the spread of antibiotic-resistant *S. aureus* and of heat-resistant staphylococcal enterotoxins [45]. The presence of MRSA strains in RTE fresh produce may pose a further threat to public health, and therefore, the use of EOs as natural disinfectants can improve the safety of these foods. Last, regarding the effectiveness of the EO mixture, a synergistic effect against all tested strains was observed. The EO/EO combination led to a significant cell viable count decrease in all tested strains, with a reduction of 2.5 log cfu/g (*p* = 0.0007) in artificially contaminated RTE fruits with *S. typhimurium* ATCC 19585 after 24 h of exposure. This increased activity was maintained throughout the experiment, with a decrease in viable cells of the tested bacteria of at least 7 logs after 8 days. Regarding *E. coli* and *L. monocytogenes*, the decrease in viable cells was 7.7 log at the end of the experiment. The synergy between cinnamon EO/clove EO against food-borne pathogens and spoilage bacteria (*S. aureus*, *L. monocytogenes*, *S. typhimurium* and *P. aeruginosa*) was also reported by Purkait et al. [46]. The EO/EO combination led to an increase in the antibacterial effect, with a decrease in their employed concentrations. An important aspect to evaluate, before any use of essential oils as preservatives, is the alteration of flavor. The olfactory contribution that even small amounts of these compounds bring to food could be remarkable, and not all consumers are sure to like it. Such a negative aspect can be overcome by combining two essential oils, as also revealed in the present investigation. In our “on food” tests, we mixed essential oils derived from different plants, and the encouraging results obtained support the possibility of their use in fresh-cut fruit, similar to what happens for vegetables, where these natural additives are already used as flavors. Even a careful selection of EOs, based on food taste characteristics, could contribute to obtaining a minimal impact on the organoleptic properties of the product. Last, seeing the rapid emergence of drug-resistant pathogens, the antimicrobial efficacy of EOs could be exploited in depth. The emergence of antibiotic-resistant pathogens in the food chain is considered a cross-sectoral problem, and fresh-cut fruits could also represent a favorable environment for exchange through the conjugation mechanism of genes responsible for antibiotic resistance and a vehicle of difficult-to-treat infections. As revealed in other investigations, EOs are natural substances capable of positively modulating the sensitivity of antibiotic-resistant pathogens [47], even when they are organized in biofilms [48]. Thus, from today’s perspective of One Health approaches to infectious diseases, EOs also represent a valid alternative as a means of dealing with the problem of antibiotic-resistant pathogens. 

## 4. Materials and Methods

### 4.1. Microbial Strains and Essential Oils

Five food pathogen classified strains were used in this study, including *Yersinia enterocolitica* ATCC 23715, *Escherichia coli* ATCC 25922, *Salmonella typhimurium* ATCC 19585, *Listeria monocytogenes* NCTC 10888 and *Staphylococcus aureus* ATCC 6538.

All strains were confirmed by matrix-assisted laser desorption ionization (MALDI) time-of-flight mass spectrometry (TOF/MS) and maintained in Tryptic Soy Broth (TSB, Oxoid S.p.A, Milan, Italy) supplemented with 20% (*vol.*/*vol.*) glycerine at −80 °C until use.

Cinnamon (*Cinnamomum zeylanicum*) and clove (*Syzygium aromaticum* L. Merr. & Perry) essential oils (EOs) were purchased from a local herbalist shop. EOs were stored at a low temperature (4 °C) and protected from light and humidity until use. 

### 4.2. Chemical Characterization of the EOs

#### 4.2.1. Gas Chromatography–Mass Spectrometry (GC–MS) Analysis

Analyses were performed on a 7890A gas chromatograph coupled with a 5975C network mass spectrometer (GC–MS) (Agilent Technologies, Milan, Italy). Compounds were separated on an Agilent Technologies HP-5 MS cross-linked poly-5% diphenyl–95% dimethyl polysiloxane (30 m × 0.25 mm i.d., 0.25 μm film thickness) capillary column. The column temperature was initially set at 45 °C, increased at a rate of 2 °C/min up to 100 °C, raised to 250 °C at a rate of 5 °C/min and finally held for 5 min. The injection volume was 0.1 μL with a split ratio of 1:20. Helium was used as the carrier gas at a flow rate of 0.7 mL/min. The injector, transfer line and ion source temperatures were 250, 280 and 230 °C, respectively. MS detection was performed with electron ionization at 70 eV, operating in the full-scan acquisition mode in the *m/z* range 40–400. The EOs were diluted 1:20 (*v*/*v*) with *n*-hexane before GC–MS analysis. 

#### 4.2.2. Gas Chromatography–Flame Ionization Detection (GC–FID) Analysis

Chromatographic characterization of EOs was performed on a 7820 gas chromatograph (Agilent Technologies, Milan, Italy) with a flame ionization detector (FID). EOs and the mixture of aliphatic hydrocarbons (C_8_–C_40_) were diluted 1:20 (*v*/*v*) with *n*-hexane before GC–FID analysis. Helium was used as a carrier gas at a flow rate of 1 mL/min. The injector and detector temperatures were set at 250 and 300 °C, respectively. EO components were separated on an Agilent Technologies HP-5 crosslinked poly-5% diphenyl–95% dimethylsiloxane (30 m × 0.32 mm i.d., 0.25 μm film thickness) capillary column. The column temperature was initially set at 45 °C, increased at a rate of 2 °C/min up to 100 °C, raised to 250 °C at a rate of 5 °C/min and finally maintained for 5 min. The injection volume was 1 μL with a split ratio of 1:20. 

Compounds were identified by comparing the retention times of the chromatographic peaks with those of authentic reference standards run under the same conditions and by comparing the linear retention indices (*LRI*s) relative to C_8_–C_40_ n-alkanes obtained on the HP-5 column under the above-mentioned conditions according to the literature [49]. Peak enrichment by co-injection with authentic reference compounds was also carried out. A comparison of the MS fragmentation pattern of the target analytes with those of pure components was performed by using the National Institute of Standards and Technology (NIST version 2.0d, 2005) mass spectral database. 

The percentage of the relative number of individual components was expressed as the percent peak area relative to the total peak area obtained by the GC–FID analysis. Semi-quantitative data were acquired from the mean of two analyses. 

Data acquisition and processing were performed using OpenLab CDS C.01.04 (Agilent Technologies, Santa Clara, CA, USA) software.

### 4.3. Agar Disk Diffusion Assay

The preliminary determination of the antibacterial activity of cinnamon and clove EOs against all tested strains was carried out by using the agar disk diffusion assay according to the standard procedure of the Clinical and Laboratory Standards Institute [50]. Sterile disks of 6 mm in diameter containing 10 µL of each EO were placed on Tryptic Soy Agar (TSA, Oxoid S.p.A, Milan, Italy) plates previously seeded with 100 µL of 10^6^ CFU/mL of each cell suspension. After incubation at 37 °C for 24 h, the antagonistic activity of the EOs was quantified by a clear zone of inhibition of the bacterial growth around the disks, and the diameters in millimeters of these zones were measured [51]. 

### 4.4. Minimum Inhibitory Concentration (MIC)

According to the Clinical Laboratory Standards Institute (CLSI) guidelines (2019) [52], the MIC of EOs was determined against all microorganisms using the broth microdilution method in 96-well microplates. Briefly, in each well of a sterile 96-well microplate, 95 µL of Tryptic Soy Broth (TSB, Oxoid S.p.A, Milan, Italy) and 5 µL of bacterial suspension were added to a final inoculum concentration of 10^6^ CFU/mL. Then, 100 µL of EO serial dilutions was added to obtain concentrations ranging from 512 to 0.125 μg/mL. Negative control wells consisted of bacteria in TSB without EOs. The plates were incubated at 37 °C for 24 h on a plate shaker at 150 rpm. The MIC was defined as the lowest concentration of EOs that inhibited visible growth of the tested microorganisms when the optical density (OD) was measured at 570 nm using a microtiter plate reader. All experiments were conducted in triplicate, and the results were expressed as the arithmetic mean of the three determinations.

### 4.5. Determination of the Fractional Inhibitory Concentration Index (FIC Index)

The checkerboard method [53] was carried out to check the synergistic antibacterial activity of the combined EO/EO by using the microdilution method in the same way as previously described for the MIC evaluation. The FIC index value was calculated by comparing the value of the MIC of each EO alone with the combination-derived MIC. An FIC index value of ≤0.5 reveals synergism, ≤0.5 to ≥1 reveals an additive effect, 1 to 4 reveals indifference, and >4 reveals antagonism. 

### 4.6. Growth Kinetics Study 

The growth of all test strains was determined in the presence of single EOs and the EO/EO (cinnamon–clove) combination added at the MIC and FIC index values. In a 96-well sterile microplate, 100 μL of sterile nutrient broth was mixed with 50 μL of single EOs or the EO/EO combination and with 50 μL of each microbial strain from a stock previously diluted to obtain a bacteria cell density of about 10^5^ CFU/mL. Measurements were obtained in an automatic micro plate reader (Tecan Sunrise™) at an optical density (OD) of 595 nm with orbital shaking at 150 rpm for a total of 26 h at 1-h intervals. The experiments were replicated three times, and the results were expressed as the arithmetic mean of the three determinations.

### 4.7. Evaluation of the Antibacterial Activity by “On Food” Studies

The antibacterial activity of the EOs, both individually and in combination, was assessed in individually packaged samples of fresh-cut fruits (a mixed pool of watermelons, pineapples, strawberries and peaches) purchased from a local supermarket on the first day of their shelf life (indicated by the expiration date on the package). Before starting the study, the microbial contamination of the fresh-cut fruits was determined. On average, the samples exhibited a microbial load of 18 CFU/g. The presence of bacteria used in this study was excluded. 

The samples were artificially contaminated by a 10 min immersion in 50 mL of each bacterial suspension (10^8^ CFU/mL) in sterile saline solution (NaCl 0.85%). After drying, the contaminated samples were submerged for 10 min in 20 mL of a solution (sterile water and sucrose 1%) of single EOs and the EO/EO combination (EO/EO), added to the values of the MIC and FIC index. Contaminated samples without immersion in a solution containing EOs were used as controls. 

Subsequently, portions of fresh-cut fruit were packed in 10 g servings and stored at refrigeration temperature (4 °C). At regular intervals (0 h, 24 h, 4 days, 6 days and 8 days), the viable load of each strain was determined by direct counting in selective plates. Individual portions (10 g) were opened and collected in sterile plastic bags, supplemented with 90 mL of Buffered Peptone Water (Oxoid, Milan, Italy) and homogenized for 1 min in Stomacher (Stomacher Lab Blender, Seward Medical, London, UK). Serial tenfold dilutions of the obtained suspensions were spread in triplicate on selective plates and incubated aerobically at 37 °C for 48 h. Viable cells of tested bacteria were enumerated, and the results were expressed as CFU/g. The experiments were replicated three times, and the results were expressed as the arithmetic mean of the three determinations.

### 4.8. Statistical Analysis

The statistical significance was determined by a *t*-test and an ANOVA using statistical program GraphPad Prism 9.2.0. (San Diego, CA, USA). The analysis was followed by Bonferroni’s post hoc test. The statistical analysis of kinetic data was performed following the “GraphPad guide to comparing dose–response or kinetic curves” [54]. For each kinetic curve obtained in this study, the Area Under the Curve (AUC) was calculated to summarize the curve into a single value. The statistical analysis was performed on the AUC values of each experimental group using an unpaired *t*-test. The *p*-values were declared significant at ≤0.05. To verify the reproducibility of the results, each experiment was replicated three times under the same conditions. 

## 5. Conclusions

The results of the present study provide encouraging information concerning the effects of two natural antimicrobial agents on the safety of minimally processed fruits. The results also highlight the importance of the synergistic effect of the EO/EO combination based on the FIC index, time–kill assay and “on food” studies. This synergistic effect is shown by the efficacy in inhibiting the growth and survival of pathogenic bacteria at low concentrations, even when applied in fresh-cut produce. Thus, the application of this EO mixture in fruit produce would also be acceptable to consumers not only for its microbial safety obtained with the addition of natural preservatives like EOs but also for its organoleptic characteristics, which are not altered by the low concentrations of use. These encouraging data can suggest the EO mixture as a new antimicrobial strategy for the correct preservation of perishable products like fresh-cut fruits. 

To confirm the effectiveness of the low EO concentrations of the synergistic mixture, further studies will be performed on its ability to damage the structural integrity of the cell membrane of the pathogenic strains, both in planktonic and in biofilm states. Studies on the toxicity, safety and interaction of EOs at the cellular level still need an in-depth investigation.

## Figures and Tables

**Figure 1 antibiotics-13-00319-f001:**
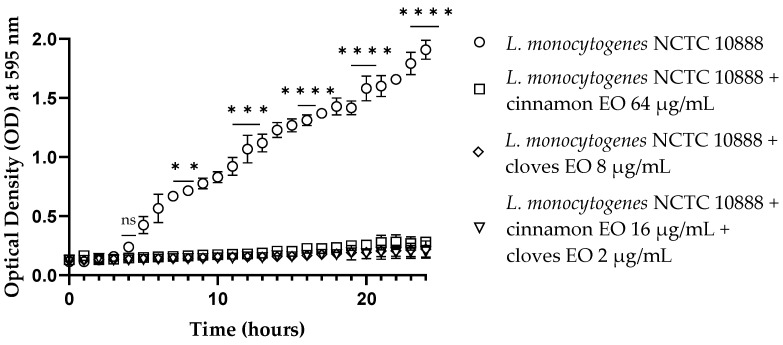
Time–kill studies of cinnamon and clove EOs alone and in combination against *Listeria monocytogenes* NCTC 10888 viable cells. *p*-values of <0.01 (**), *p* < 0.001 (***) and *p* < 0.0001 (****) were considered significant by *t*-test and ANOVA with Bonferroni correction. Results are expressed as mean ± SD of the three determinations (error bar = S.D.; *n* = 3).

**Figure 2 antibiotics-13-00319-f002:**
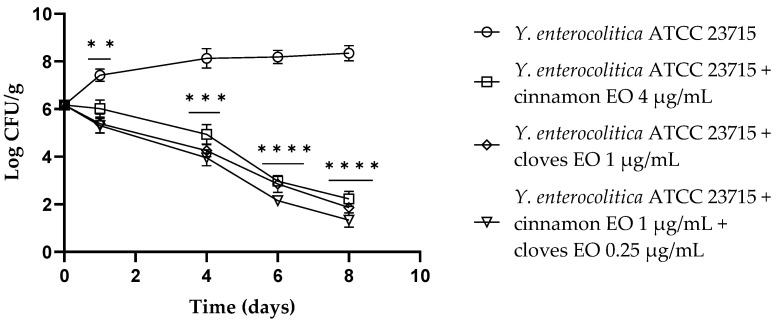
*Yersinia enterocolitica* viable counts (log CFU/g) observed in the contaminated fresh-cut fruits. *p*-values of <0.01 (**), *p* < 0.001 (***) and *p* < 0.0001 (****) were considered significant by *t*-test and ANOVA with Bonferroni correction. ns stands for not statistically significant. Results are expressed as mean ± SD of the three determinations (error bar = S.D.; *n* = 3).

**Figure 3 antibiotics-13-00319-f003:**
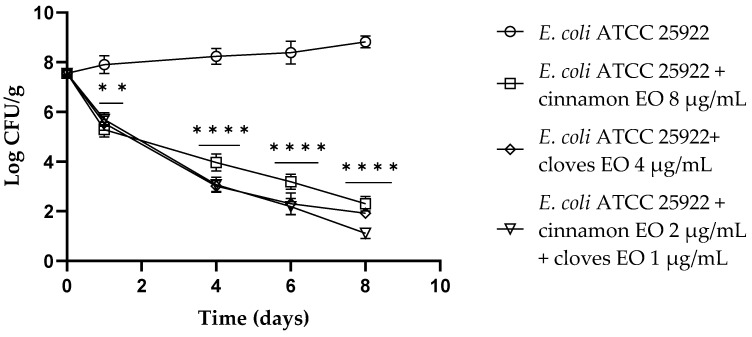
*Escherichia coli* viable counts (log CFU/g) observed in the contaminated fresh-cut fruits. *p*-values of <0.01 (**) and *p* < 0.0001 (****) were considered significant by *t*-test and ANOVA with Bonferroni correction. ns stands for not statistically significant. Results are expressed as mean ± SD of the three determinations (error bar = S.D.; *n* = 3).

**Figure 4 antibiotics-13-00319-f004:**
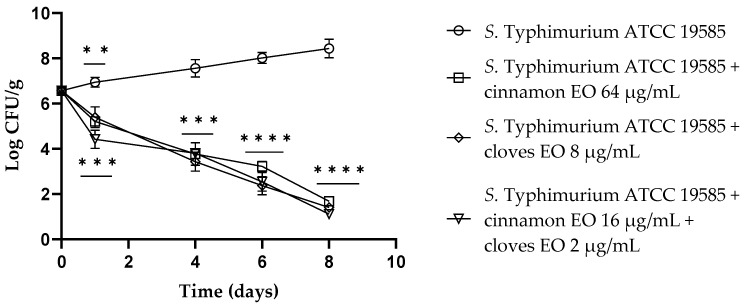
*Salmonella* Typhimurium ATCC 19585 viable counts (log CFU/g) observed in the contaminated fresh-cut fruits. *p*-values of <0.01 (**), *p* < 0.001 (***) and *p* < 0.0001 (****) were considered significant by *t*-test and ANOVA with Bonferroni correction. ns stands for not statistically significant. Results are expressed as mean ± SD of the three determinations (error bar = S.D.; *n* = 3).

**Figure 5 antibiotics-13-00319-f005:**
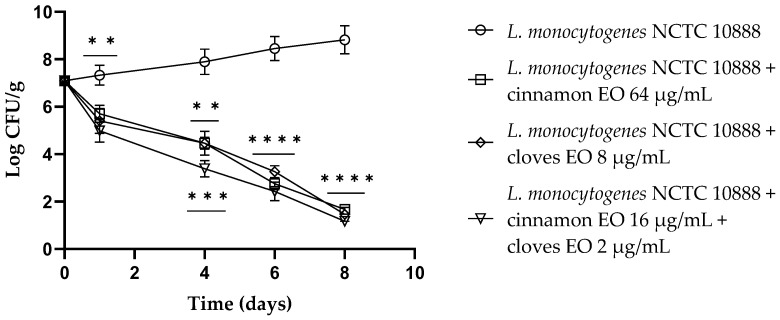
*Listeria monocytogenes* NCTC 10888 viable counts (log CFU/g) observed in the contaminated fresh-cut fruits. *p*-values of <0.01 (**), *p* < 0.001 (***) and *p* < 0.0001 (****) were considered significant by *t*-test and ANOVA with Bonferroni correction. ns stands for not statistically significant. Results are expressed as mean ± SD of the three determinations (error bar = S.D.; *n* = 3).

**Figure 6 antibiotics-13-00319-f006:**
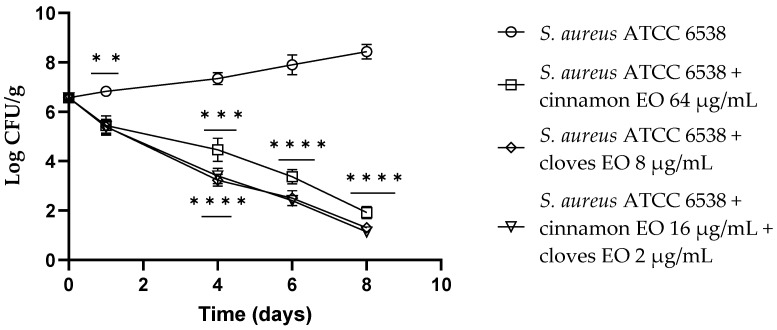
*Staphylococcus aureus* viable counts (log CFU/g) observed in the contaminated fresh-cut fruits. *p*-values of <0.01 (**), *p* < 0.001 (***) and *p* < 0.0001 (****) were considered significant by *t*-test and ANOVA with Bonferroni correction. ns stands for not statistically significant. Results are expressed as mean ± SD of the three determinations (error bar = S.D.; *n* = 3).

**Table 1 antibiotics-13-00319-t001:** Chemical percent composition of *Cinnamomum zeylanicum* (cinnamon) bark essential oil and *Syzygium aromaticum* (clove) flower bud essential oil.

*Compound*	Lit. LRI	Exp. LRI	*Cinnamomum zeylanicum*	*Syzygium aromaticum*
α-thujene	928	926	0.10	-
α-pinene	936	932	0.47	-
camphene	950	947	0.19	-
sabinene	973	972	0.21	-
α-phellandrene	1004	1002	0.37	-
α-terpinene	1017	1014	0.18	-
*p*-cymene	1024	1022	1.44	-
limonene	1029	1026	2.23	0.26
linalool	1099	1100	3.36	0.71
fenchol	1112	1116	-	0.14
camphor	1143	1143	-	0.24
terpinen-4-ol	1177	1176	0.21	-
α-terpineol	1190	1190	0.42	-
*trans*-cinnamaldehyde	1277	1278	68.96	-
bornyl acetate	1287	1287	0.30	-
thymol	1292	1292	0.13	-
citronellyl acetate	1357	1362	3.37	-
eugenol	1378	1378	0.74	79.61
α-copaene	1376	1381	-	0.34
β-caryophyllene	1420	1424	5.64	3.05
α-bergamotene	1444	1445	0.26	-
cinnamyl acetate	1445	1449	2.07	-
α-humulene	1453	1458	1.15	0.51
γ-cadinene	1523	1522	1.95	-
δ-cadinene	1527	1527	0.19	-
eugenyl acetate	1526	1535	-	11.47
caryophyllene oxide	1590	1594	1.06	1.64
Total			95.00	97.97

Experimental retention indices and literature retention indices (HP-5 column) according to NIST 14 (National Institute of Standards and Technology, USA; 14th edition) library database [29].

**Table 2 antibiotics-13-00319-t002:** Agar disk diffusion assay of cinnamon and clove essential oils. The inhibition zones were measured in mm.

Indicator Strains	Cinnamon Essential Oil	Clove Essential Oil
*Yersinia enterocolitica* ATCC 23715	32 ± 1.1	35 ± 0.6
*Escherichia coli* ATCC 25922	29 ± 1.0	31 ± 1.4
*Salmonella* Typhimurium ATCC 19585	27 ± 0.9	30 ± 1.3
*Listeria monocytogenes* NCTC 10888	29 ± 0.5	32 ± 1.6
*Staphylococcus aureus* ATCC 6538	27 ± 0.9	31 ± 1.6

**Table 3 antibiotics-13-00319-t003:** Synergistic activity of EO/EO combination, by FIC index calculation. MIC values are expressed in μg/mL.

Strains	EO	MIC EOs (μg/mL)	MIC EO/EO (μg/mL)	FIC Index
*Yersinia enterocolitica* ATCC 23715	Cinnamon Clove	4 1	1 0.25	0.5
*Escherichia coli* ATCC 25922	Cinnamon Clove	8 4	2 1	0.5
*Salmonella* Typhimurium ATCC 19585	Cinnamon Clove	64 8	16 2	0.5
*Listeria monocytogenes* NCTC 10888	Cinnamon Clove	64 8	16 2	0.5
*Staphylococcus aureus* ATCC 6538	Cinnamon Clove	64 8	16 2	0.5

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
