# Peer review of "Efficacy and Synergistic Potential of Cinnamon (Cinnamomum zeylanicum) and Clove (Syzygium aromaticum L. Merr. & Perry) Essential Oils to Control Food-Borne Pathogens in Fresh-Cut Fruits"

_antibiotics, 2024, doi:10.3390/antibiotics13040319_

Round 1

Reviewer 1 Report

Comments and Suggestions for Authors

My comments are:

- Line 35, it should say: "...fresh products plays an important role..."

- lines 38, 52, 68, 69, and 107, change the word "produce" (misspelled?) to "products"

- line 125: remove "cinnamaldehyde" (repeated). When the word "cinnamaldehyde" is used without specifying whether it is the cis- or trans-isomer, it is assumed that it refers to the more stable trans-isomer that is correctly mentioned immediately after. 

- lines 157 and 158: In Table 1, the part of the plant used to obtain the essential oil must be specified. 

- line 207, it should say: "expressed"

line 482: the authors should indicate what solvent was used to dissolve the essential oils and EO/EO combination. 

Author Response

The Authors thanks the Reviewer for helpful comments. You will find all the answers and suggested changes in the attached file Response to Review 1

Reviewer 2 Report

Comments and Suggestions for Authors

Some thing need to be revised

Author Response

The Authors thank the Reviewer for helpful comments. You will find all the answer and suggested changes in the attached file Response to Reviewer 2
